# Advances in Genome Editing Through Haploid Induction Systems

**DOI:** 10.3390/ijms26104779

**Published:** 2025-05-16

**Authors:** Huajin Sheng, Peng Gao, Changye Yang, Teagen D. Quilichini, Leon V. Kochian, Raju Datla, Daoquan Xiang

**Affiliations:** 1Department of Biology, College of Arts and Science, University of Saskatchewan, Saskatoon, SK S7N 5E2, Canada; huajin.sheng@usask.ca (H.S.); teagen.quilichini@usask.ca (T.D.Q.); 2Agriculture and Agri-Food Canada, Saskatoon Research and Development Centre, 107 Science Place, Saskatoon, SK S7N 0X2, Canada; peng.gao@agr.gc.ca; 3Aquatic and Crop Resource Development, National Research Council Canada, Saskatoon, SK S7N 0W9, Canada; changye.yang@nrc-cnrc.gc.ca; 4Department of Plant Sciences, University of Saskatchewan, Saskatoon, SK S7N 5A8, Canada; leon.kochian@usask.ca; 5Global Institute for Food Security, University of Saskatchewan, Saskatoon, SK S7N 4J8, Canada; raju.datla@gifs.ca

**Keywords:** gene editing, haploid induction, double haploid, maternal haploid inducer, paternal haploid inducer

## Abstract

Groundbreaking advances in gene editing technologies are transforming modern plant breeding by enabling precise genetic modifications that dramatically accelerate crop improvement. Haploid and diploid induction systems have emerged as particularly powerful tools in this landscape, offering both efficient gene editing capabilities and rapid production of homozygous lines while seamlessly integrating with the advanced genome-editing platforms such as CRISPR-Cas systems. This review synthesizes the current state of knowledge regarding the mechanisms, applications, and recent progress in haploid and diploid induction systems for gene editing. We examine their transformative potential for enhancing genetic gains and compressing breeding timelines, with significant implications for global food security. Additionally, we provide a critical analysis of emerging challenges of genome editing in crops and outline promising future directions for research and development.

## 1. Introduction

The increasing global population, coupled with the adverse effects of climate change on crop production, is driving a growing demand for food security and necessitating the development of improved crop varieties with enhanced yield, stress resistance, and nutritional quality [1]. Traditional crop breeding approaches, which incorporate natural genetic variation into elite germplasm through controlled crosses, have significantly advanced the improvement of major crops over recent decades [2]. Despite their effectiveness, conventional breeding methods are inherently time-consuming and labor-intensive, typically requiring a decade or more to develop a new crop variety due to the multiple rounds of crossing, selection, and field evaluation [3]. These traditional approaches are further constrained by linkage drag, where beneficial traits remain physically connected to unfavorable ones on the chromosome, making their separation challenging or impossible due to either the need for extensive breeding populations or the absence of genetic recombination in certain genomic regions [4]. Consequently, there is a pressing need to develop more efficient strategies for creating new cultivars that successfully combine multiple desired traits in major crop species.

Doubled haploid (DH) technology, leveraging haploid induction, has emerged as a powerful tool to accelerate the development of new crop varieties with desired traits. The process involves generating haploid plants containing a single set of parental chromosomes through either in vitro or in vivo methods, followed by chromosome doubling using compounds such as colchicine to produce homozygous diploid plants. These double haploid lines can then be directly released as cultivars or utilized as parent lines for hybrid seed production [5]. While traditional plant breeding methods typically require 4–6 successive generations of crossing and selection to stabilize genetic backgrounds and achieve near-complete homozygosity, a process that can span more than a decade depending on the crop and breeding strategy, DH technology offers a dramatically more efficient approach by producing completely homozygous lines in just one generation [6]. Given its ability to rapidly produce completely homozygous lines, DH technology has become a widely adopted tool in crop improvement programs, particularly in the development of parental lines for hybrid seed production [7]. Currently, DH production protocols have been established for approximately 400 species [5], resulting in over 300 DH-derived cultivars across multiple crop species worldwide [8]. The impact of this technology is particularly evident in Canada, where DH-derived wheat varieties occupy more than one-third of the country’s wheat acreage [9].

The rapid emergence and advancement of modern biotechnological and genomic tools, including DNA sequencing techniques, marker-assisted selection, genomic selection, plant transgenic methods, and genome editing, have dramatically enhanced crop breeding efficiency [10]. Among these powerful tools, genome editing distinguishes itself through its precision and versatility, enabling targeted modifications of specific genes for rapid improvement of agronomic traits while significantly reducing costs. The fundamental mechanism of genome editing involves two key steps: first, the recognition of target genomic loci and binding of an effector DNA-binding domain (DBD), followed by the introduction of double-strand breaks (DSBs) in the target DNA by restriction endonucleases [11]. These DSBs are subsequently repaired through the cell’s endogenous DNA repair pathways, either via homology-directed recombination (HDR) or non-homologous end joining (NHEJ) [12]. Several major genome-editing tools have been developed, including homing endonucleases or meganucleases (HEs) [13], zinc finger nucleases (ZFNs) [14], transcription activator-like effector nucleases (TALENs) [15], and clustered regularly interspaced short palindromic repeats (CRISPR) with CRISPR-associated protein (Cas) [16,17]. The CRISPR/Cas system, which utilizes base complementarity between single-guide RNA (sgRNA) and target DNA for recognition, has revolutionized gene editing by simplifying the cloning process. Its exceptional attributes—simplicity, precision, adaptability to various targets, and capacity for multiplexed editing—have established it as the preeminent choice for engineering crop traits [1].

CRISPR-Cas systems are organized into two classes (Class 1 and Class 2), six types (I to VI), and numerous subtypes. Class 1 systems (Types I, III, and IV) utilize multi-Cas protein effector complexes, while Class 2 systems (Types II, V, and VI) function with a single effector protein. Among these, the Type II CRISPR-Cas9 system from Streptococcus pyogenes (SpCas9) has been extensively characterized and widely adopted. Following the pioneering applications of CRISPR/Cas technology in plant cells [18,19,20], CRISPR/Cas-mediated genome editing has been successfully implemented across diverse crop species to elucidate gene functions and develop improved varieties with enhanced yield, stress tolerance, nutrient efficiency, and nutritional quality [21,22]. Continuous innovations are optimizing CRISPR/Cas systems in plants to address key limitations, including off-target effects, inefficient delivery of editing tools, restrictive protospacer adjacent motif (PAM) sequences, and low frequency of targeted gene insertion or replacement through HDR [23,24]. Nevertheless, significant challenges remain in genome editing for crop improvement. The process requires plant transformation, involving the delivery of editing reagents into plant cells, selection of edited cells, and regeneration of intact plants with desired modifications [25]. A major obstacle is the application of genome editing to crop species and elite varieties that are resistant to established transformation methods [26]. Consequently, targeted mutations must first be generated in transformation-amenable lines and then introgressed into elite germplasm through backcrossing—a process that is challenging, expensive, and time-consuming. Another significant challenge is the creation of homozygous and transgene-free plants to ensure trait stability and meet regulatory requirements. At the same time, various approaches exist—including genetic segregation during sexual reproduction, marker-assisted tracking, programmed self-elimination of transgenes, and non-transgenic methods such as ribonucleoprotein transfection [27]. However, these strategies often require additional generations and larger populations, particularly for crops with complex polyploid genomes or when targeting multiple genes.

The integration of genome editing with DH technology offers a transformative approach to overcome the challenges associated with both technologies in crop improvement. Following the elucidation of molecular and genetic mechanisms underlying haploid induction, CRISPR/Cas-based genome editing targeting haploid induction genes has emerged as a precise and powerful tool for developing haploid inducer lines across various crops [28,29]. Additionally, direct editing of gametes or haploid embryos provides significant advantages over conventional plant transformation systems, including efficiency and reduced generation time. This review examines recent advances in three key areas: the development of haploid inducer lines through genome editing, the direct introduction of edits in haploid tissues, and the synergistic application of DH and genome editing technologies for hybrid breeding. We also analyze current technical limitations, including efficiency variations across crop species, and explore future directions for optimizing these technologies to accelerate crop improvement.

## 2. Haploid Induction Systems in Crop Breeding

DH technology enables the efficient production of completely homozygous lines from heterozygous donor plants in a single step [6]. Since the initial discovery of a haploid mutant in the weed species *Datura stramonium* L. [30] and the subsequent first report of haploids from in vitro anther culture of *Datura innoxia* [31], significant advances in haploid techniques have facilitated the widespread adoption of DH technology in plant breeding. Today, DH technology has become integral to advanced breeding programs across numerous crop species, including maize (*Zea mays*), barley (*Hordeum vulgare*), *Brassica* sp., wheat (*Triticum aestivum*), and various vegetable crops [32,33,34].

DH production involves two primary steps: haploid induction and chromosome doubling (Figure 1A). Haploid induction generates plants with a single set of chromosomes (n) derived from the donor plant [35]. This can be achieved through androgenesis, gynogenesis, or parthenogenesis using either in vitro or in vivo methods, depending on the species [36,37,38,39]. In vitro methods rely on culturing haploid gametophytic cells to produce haploid embryos and plants but are highly genotype-dependent and influenced by donor plant condition, pollen stage, anther pretreatment, and culture conditions, limiting their broader use in breeding programs [40].

The in vivo haploid induction system provides promising alternatives to overcome the efficiency and time constraints inherent to in vitro systems. These methods utilize specific pollination techniques, including irradiated pollen, interspecific crosses (wide cross), or intraspecific crosses with inducer lines [35,41,42,43,44,45]. The in vivo haploid induction system, facilitated by haploid inducer (HI) lines, represents a unique in planta method that eliminates tissue culture requirements (Figure 1A). Fertilization with an HI plant can produce a haploid embryo either through single fertilization (sperm cell fusion with the central cell) or through post-zygote genome elimination in double fertilization [46]. This approach enables haploid embryo and plantlet production through simple intra-specific crosses and is regarded as the most effective method for obtaining DHs [29]. However, haploid inducer lines are currently routinely employed only in maize breeding. The first maize haploid inducer line, Stock6, capable of inducing haploid embryo development in target maize lines upon pollination with its pollen, was discovered in 1959 [47]. Subsequently, a series of high-efficiency haploid inducer lines derived from Stock6 have been developed and extensively utilized in maize breeding programs [48].

## 3. Haploid Genes Discovery and Genome Editing Applications for Haploid Inducer Line Development

In vivo haploid induction triggered by haploid inducer lines operates through two distinct pathways: maternal and paternal haploid induction, determined by their respective roles in the cross. Among these, maternal haploid induction has received the most research attention and has found widespread application in maize breeding programs [29,49]. Genetic studies utilizing haploid inducer lines have identified two major QTLs, *qhir1* and *qhir8*, as key determinants of maternal haploid induction in maize. Through successive rounds of fine mapping, the first critical gene controlling maternal haploid formation, designated as *MATL*/*NLD*/*ZmPLA1*, was identified and functionally validated through genome editing [46,48,50]. *MATL*/*NLD*/*ZmPLA1* is a pollen-specific gene encoding a predicted patatin-like phospholipase A, with most maize haploid inducer lines carrying a 4 bp insertion at the end of the coding sequence [48]. Functional validation through both CRISPR/Cas9-mediated genome editing and TALEN technology confirmed that knockout or mutation of this phospholipase gene in non-inducer backgrounds leads to average haploid induction rates (HIR) of 2–6.65% (Table 1). The biochemical and molecular mechanisms underlying this haploid induction process have been elucidated through detailed molecular and cellular analyses. The *MATL*/*NLD*/*ZmPLA1* mutation causes a frameshift, resulting in a truncated protein that exhibits both mislocalization and instability. Under normal conditions, functional *MATL/NLD*/*ZmPLA1* enzyme catalyzes the hydrolysis of phospholipids, particularly phosphatidylcholine, in sperm cell membranes. This activity maintains phosphatidylcholine at homeostatic levels in wild-type pollen. However, loss of functional *MATL*/*NLD*/*ZmPLA1* in mutant pollen disrupts this homeostasis, leading to significant accumulation of phosphatidylcholine in sperm cell membranes [51]. This phospholipid accumulation triggers a cascade of downstream effects, most notably the production of reactive oxygen species (ROS). Elevated ROS levels create an oxidative environment that damages sperm DNA integrity, resulting in extensive DNA fragmentation. Remarkably, this DNA damage does not prevent double fertilization, but instead leads to post-fertilization elimination of the damaged paternal genome from the zygote. Through this selective genome elimination process, haploid seeds containing only the maternal genome are generated [51]. Understanding this biochemical pathway provides several opportunities for optimizing haploid induction efficiency. First, targeted approaches to enhance phosphatidylcholine accumulation in sperm cells, either through additional modifications to phospholipid metabolism genes or through chemical treatments that inhibit phosphatidylcholine catabolism, could potentially increase HIR. Second, strategies aimed at amplifying ROS production or stability in sperm cells might enhance DNA fragmentation efficiency and subsequent genome elimination. Finally, engineering modifications that affect the timing of DNA damage or genome elimination during early embryogenesis could potentially increase the proportion of successful haploid embryo formation.

The conservation of *MATL*/*NLD*/*ZmPLA1* across cereal plants, coupled with the discovery of *ZmPLA1* in maize, has catalyzed the expansion of haploid induction to other crops through genome editing technology. In developing a haploid inducer for rice (Oryza sativa), researchers identified *OspPLAIIφ* (*Os03g27610* (*PLP1*)) as the putative *ZmMATL* orthologue from among 16 patatin-like phospholipases genes, based on its pollen-specific expression pattern and sequence similarity to *ZmMATL*. Using CRISPR/Cas9 technology, researchers generated a series of knockout mutations. Subsequent ploidy analysis revealed that the average HIR for *Osmatl* mutants was approximately 6%, comparable to the 6.7% rate observed in C-terminal-edited matl lines in maize [52]. The wheat genome harbors three homologous *MTL* genes (*TraesCS4A02G018100*, *TraesCS4B02G286000*, and *TraesCS4D02G284700*), sharing approximately 70–80% amino acid sequence identity with *ZmMATL*. Individual knockout of *TaMTL-A* yields average HIRs ranging from 5.88% to 15.66% [53]. A recent study utilizing an optimized CRISPR/SpCas9 system to edit wheat *TaMTLs* demonstrated that double-knockout mutations of *TaMTL-4A* and *TaMTL-4D* produced a 10% haploid induction frequency, while triple-knockout of *TaMTL-4A*, *TaMTL-4B*, and *TaMTL-4D* achieved frequencies of 11.8–31.6% [54]. Extending this approach to other cereals, CRISPR/Cas9-mediated knockout of *SiMTL* in foxtail millet and *HvMTL* in barley generated HIRs of 2–3% and 11–12%, respectively [55,56]. Most recently, researchers successfully established an in vivo maternal haploid induction system in sugarcane by editing the *MTL*/*PLA1*/*NLD* homolog *ScMTL*, achieving HIRs of 0.59–0.96% [57]. However, the absence of functional *MTL* orthologs in dicot species has confined the application of this haploid induction system primarily to monocot species.

The *DMP* (DUF679 membrane protein) gene family represents a fascinating example of evolutionary conservation combined with functional adaptation across plant lineages. Unlike *MTL*/*NLD*/*ZmPLA1* genes, which are primarily conserved in monocots, *DMP* genes show remarkable conservation in both monocot and dicot species. This phylogenetic distribution suggests that *DMP* proteins likely evolved before the monocot-dicot divergence approximately 150–200 million years ago and have maintained essential functions throughout angiosperm evolution. A critical analysis of *DMP*-mediated haploid induction reveals significant variation in efficiency across species. In maize, *ZmDMP* knockout alone produces minimal HIR (0.1–0.3%), functioning primarily as an enhancer when combined with matl/nld/zmpla1 mutations, where it amplifies HIR by 5–6 fold [58]. In contrast, Arabidopsis *dmp8dmp9* double mutants achieve a relatively higher average HIR of 2.1% [59]. In *Medicago truncatula*, simultaneous CRISPR/Cas9 mutation of *MtDMP8* and *MtDMP9*, which share the highest similarity with *ZmDMP*, triggered in planta haploid induction with 0.29–0.82% maternal HIR [60]. The tomato genome features a single DMP gene (*SlDMP*; Solyc05g007920) with high expression in pollen and flower buds. Its knockout resulted in 1.8% HIR through selfing and 0.5–3.7% through outcrossing with 36 different female genotypes, demonstrating the potential for genotype-independent haploid induction [61]. In allotetraploid *Nicotiana tabacum*, three *DMP* genes (*NtDMP1*, *NtDMP2*, and *NtDMP3*) show high similarity to *ZmDMP*, *AtDMP8*, and *AtDMP9*, with strong expression in anthers during the binucleate pollen grain stage. CRISPR/Cas9-generated dmp1dmp2dmp3 triple mutants achieved maternal haploid induction rates of 1.52–1.75 [62]. The *Brassica* family presents multiple *DMP* homologs: four in *B. napus* (*BnDMP1A*, *BnDMP1C*, *BnDMP2A*, and *BnDMP2C*) and two in *B. oleracea* (*BoC04.DMP9* and *BoC03.DMP9*), all showing high similarity to *ZmDMP* and strong expression in anthers/pollen and flower buds. In *B. napus*, CRISPR/Cas9-generated *dmp* loss-of-function mutants in the Westar cultivar background showed varying HIRs: triple mutants achieved 1.0–4.4% through selfing, while outcrossing tests with double, triple, and quadruple mutants yielded average HIRs of 1.1%, 1.5%, and 2.4%, respectively. These results indicate that disrupting two *BnDMP* genes suffices to induce haploids in *B. napus* [63,64]. Similarly, successful haploid inducer lines were generated through knockout of *BoC03.DMP9* in *B. oleracea*, *ClDMP3* in watermelon, and *CsDMP* in cucumber, achieving HIRs of up to 1.60%, 1.12%, and 0.4%, respectively [65,66,67]. These efficiency differences suggest species-specific adaptations in *DMP* function and integration with reproductive pathways.

The variation in HIR across species presents an apparent paradox: why is a highly conserved gene family associated with such diverse haploid induction efficiencies? This divergence likely reflects differences in reproductive biology, genetic redundancy, and interactions with species-specific reproductive and developmental pathways. In dicots like Arabidopsis, tomato, and Brassica species, *DMP* mutations alone can induce haploids at practically useful frequencies, whereas in maize, *DMP* primarily functions as a modifier of *MTL*-mediated induction. The evolutionary conservation of *DMP* genes in both monocots and dicots, despite the absence of clear *MTL* orthologs in dicots, suggests that *DMP* genes may have evolved primordial roles in reproductive development that predate specialized haploid induction systems. This conservation pattern indicates that the reproductive functions of *DMP* genes represent a more ancient evolutionary innovation than the specialized roles of *MTL* genes in haploid induction. The fact that *DMP* genes can induce haploids in dicots lacking *MTL* orthologs further suggests that *DMP*-mediated haploid induction likely operates through mechanisms at least partially independent from phospholipase-triggered pathways. These evolutionary and functional insights provide significant opportunities for breeding applications. The broad conservation of *DMP* genes across diverse plant lineages offers a universal target for engineering haploid induction in species where other approaches have been unsuccessful. Furthermore, the additive effects observed between DMP and other haploid induction genes suggest promising avenues for pyramiding mutations to achieve commercially viable HIR across diverse crop species.

Maize *PHOSPHOLIPASE D3* (*ZmPLD3*), a member of the phospholipase D subfamily, encodes another phospholipase specifically expressed in pollen. Its significant upregulation in the *mtl*/*zmpla1*/*nld* background suggests that *ZmPLD3* may play a role analogous to *MTL*/*ZmPLA1*/*NLD* in haploid induction. CRISPR/Cas9-induced mutations of *ZmPLD3* demonstrated that knockout of this gene resulted in a HIR similar to that of *mtl*/*zmpla1*/*nld*. In addition, *zmpld3* and *mtl*/*zmpla1*/*nld* exhibited synergistic effects in enhancing the HIR, with the double mutants of *zmpld3* and *mtl*/*zmpla1*/*nld* increasing the HIR up to approximately 4% [68]. The high conservation of *ZmPLD3* in cereals suggests potential applications in other crops. However, the extension of this knowledge to other crops through genome editing has not yet been validated. Most recently, a new gene named *ZmPOD65*, which encodes a sperm-specific peroxidase, was identified as a regulator of haploid induction in maize. *zmpod65* mutants produced by CRISPR/Cas9 exhibited a HIR of up to7.7% [51]. In recent years, several maternal factors for maternal haploid induction were identified. The BABY BOOM (*BBM*) AINTEGUMENTA-LIKE (*AIL*)*AP2*/*ERF* domain transcription factor is a major regulator in early plant embryo and endosperm development. A gene encoding transcription factor Apospory-specific Genome Region BabyBoom-like (*ASGR BBML*) 1 was identified in parthenogenetic pearl millet (*Pennisetum squamulatum*) and has been reported to induce parthenogenesis in egg cells of sexual pearl millet [69]. Further studies have shown that when *PsASGR-BBML1* is ectopically expressed in maize and rice egg cells, it can induce the formation of haploid embryos in both species [69,70,71]. In Arabidopsis, KOKOPELLI (*KPL*, AT5G63720) encodes a putative ubiquitin E3 ligase, with the *kpl* mutant producing aberrant sperm cells with reduced fertility. Further studies demonstrated that inactivation of *AtKPL* triggers in planta maternal haploid induction, establishing *Atkpl* mutants as potential haploid inducer lines. The varying HIR between *Atkpl-1* (0.34%) and *Atkpl-2* (0.07%) likely stem from differences in allele strength, as evidenced by the more severely reduced seed set in *Atkpl-1* compared to *Atkpl-2*. Although the ~0.34% HIR of the *Atkpl-1* single mutant is relatively low for practical plant breeding applications, this efficiency might differ in other species. Collectively, both the evaluation of *kpl* mutants across diverse species and the exploration of interactions between *kpl* mutations and other genetic alterations highlight the significance of this newly identified player in haploid induction [72].

In *Arabidopsis*, mutation in gynoecium-expressed phospholipase AII (*pPLAIIγ*) triggers maternal haploid plants in Arabidopsis, at an average rate of 1.07%. Reciprocal crosses demonstrate that haploid plants are triggered from the female side and not from the pollen, and the haploid plants carry the maternal genome [73]. Additionally, a female in vivo haploid-induction system has been developed via simultaneous mutagenesis of two egg cell-specific aspartic endopeptidases (*ECS1* and *ECS2*), *ecs1 ecs2* double mutant can be used as a novel maternal HI line with an HIR of about 1%. This HI system has also been reported to work efficiently in rice [74].

In addition to maternal haploid induction, paternal haploids can be produced when haploid inducers serve as female parents, resulting in haploids containing the male genome but female cytoplasm [29]. The first gene responsible for paternal haploid induction is *indeterminate gametophyte1* (*ZmIG1*) from maize. *ZmIG1* was identified as a LATERAL ORGAN BOUNDARIES (*LOB*) domain gene and shares a high similarity with the *ASYMMETRIC LEAVES2* (*AS2*) gene in *Arabidopsis*. The paternal HIR of the *ig1* mutant can be as high as 8% [75].

The real breakthrough of the paternal haploid induction system development came from the *CENH3* study in Arabidopsis. CENTROMERIC HISTONE3 (*CENH3*) gene encodes a centromere-identifying protein histone H3 variant that functions in centromere formation and kinetochore stabilization. The CENH3-based haploid induction represents one of the most versatile yet technically challenging approaches for developing haploid inducer lines. Unlike maternal haploid inducers, which can often be created through simple knockout mutations, CENH3-based systems require precise, often complex modifications to the centromeric histone H3 variant. These modifications must be carefully calibrated to maintain sufficient CENH3 functionality for viability while creating centromeres that are prone to elimination during early embryogenesis. The complexity of required CENH3 modifications presents significant technical challenges. Early CENH3-based haploid induction systems relied on replacing the N-terminal domain with conventional histone tails or GFP fusions (“tailswap” approach), which yielded impressive HIRs of 25–45% in Arabidopsis [76,77]. However, similar approaches in crops like maize have produced disappointingly low HIRs (<1%) [78], highlighting the species-specific nature of these modifications. More recent approaches have focused on point mutations, in-frame deletions, or restored frameshift mutations, with variable success rates across species. This variability in efficacy demonstrates that subtle changes in *CENH3* structure can dramatically affect centromere function and haploid induction capability in species-specific ways. Complementing Arabidopsis *cenh3* mutant with unaltered *CENH3* variants from distantly related species such as *L. oleraceum* and *B. rapa* could also induce paternal haploid induction [79]. EMS-derived point mutations, single amino acid substitutions, and CRISPR/Cas9-mediated in-frame deletions in the *CENH3* were also successfully applied in creating haploid inducer lines with HIR ranging from 0.6 to 44% depending on the mutation [80,81]. Since *CENH3* is widely present in eukaryotes and exhibits high conservation across diverse organisms, the discovery of the CENH3-based HI system offers great promise for developing novel tools for creating haploid inducer lines in different plant species. CRISPR/Cas9 was applied to produce haploid inducers of carrot through targeted mutations of *CENH3* [82]. In wheat, there are two *CENH3* genes, *TaCENH3α* and *TaCENH3β*; each gene has three homeologs distributed into A, B, and D genomes. Based on the gene expression check, *TaCENH3α* has higher expression in pollen and ovule. Restored frameshift (RFS) mutation in the N-terminal region of *TaCENH3α-A* was introduced via CRISPR-Cas9-mediated genome editing. When RFS mutation was in heterozygous status, the paternal haploid induction rate reached 7–8% in the wheat line [83]. Most recently, a CENH3-based haploid induction system in *Brassica oleracea* was established using the CRISPR/Cas9 system to create in-frame deletions and restore frameshift mutants of *BoCENH3*. The HIR was 0.52% on average for in-frame deletion and 1.14% for restored frameshift mutation. The concept of CENH3-based haploid inducers also facilitates the transfer of cytoplasm between different genotypes. Haploid inducer lines with CMS cytoplasm were further generated via hybridization with the CMS line, enabling the production of homozygous CMS lines in one step [84].

The diverse range of effective CENH3 modifications represents both a challenge and an opportunity for emerging precision gene editing technologies. First-generation CRISPR/Cas9 systems, while effective for creating knockouts, lack the precision needed for many *CENH3* modifications. However, recent advances in base editing and prime editing technologies offer promising solutions. Base editors can create precise single nucleotide substitutions without requiring double-strand breaks, enabling the introduction of specific amino acid changes that affect *CENH3* functionality without comprehensive protein disruption. Prime editing systems offer even greater flexibility, potentially enabling precise insertions, deletions, and substitutions that could recreate specific *CENH3* variants proven effective in model systems. The technical requirements of *CENH3* modification also present opportunities for exploring combinatorial approaches. Recent research has demonstrated that integrating tailswap *CENH3* with other haploid induction mechanisms can synergistically enhance HIR [85]. This suggests that precision editing of *CENH3*, combined with modifications to other haploid induction genes, could potentially generate super-inducer lines with commercially viable HIRs across multiple crop species. Furthermore, CENH3-based systems offer unique advantages beyond standard haploid induction, including their ability to function in both maternal and paternal induction modes. The recent demonstration that CENH3-based haploid inducers can facilitate cytoplasm transfer between different genotypes [84] highlights additional applications in breeding programs, particularly for engineering cytoplasmic male sterility systems. These extended applications may justify the technical complexity of developing CENH3-based inducers in crops where simpler approaches yield insufficient HIRs. As precision gene editing technologies continue to evolve, the challenges of creating optimal CENH3 variants will likely diminish, while the opportunities for developing versatile, high-efficiency haploid induction systems across diverse crop species will expand. This evolution may ultimately position CENH3-based systems as the preferred approach for commercial haploid induction, despite their current technical complexity.

**Table 1 ijms-26-04779-t001:** Genome editing applications for haploid inducer line development.

Gene	Species	HIR (%)	Haploid Induction Type	Reference	Gene Editing Tool
*ZmPLA1*/*NLD*/*MTL*	*Oryza sativa*	2.8–12.0%	Maternal	Lv et al., 2023 [29]	CRISPR/Cas9
*ZmPLA1*/*NLD*/*MTL*	*Zea mays*	4.7–11.0%	Maternal	Chaikam et al., 2019 [32]	CRISPR/Cas9
*ZmPLA1*/*NLD*/*MTL*	*Zea mays*	4.0–12.5%	Maternal	Kelliher et al., 2017 [46]	TALEN
*ZmPLA1*/*NLD*/*MTL*	*Zea mays*	1.9–6.7%	Maternal	Liu et al., 2017 [48]	CRISPR/Cas9
*ZmPLA1*/*NLD*/*MTL*	*Oryza sativa*	2.0–6.0%	Maternal	Yao et al., 2018 [52]	CRISPR/Cas9
*ZmPLA1*/*NLD*/*MTL*	*Triticum aestivum*	5.9–15.7%	Maternal	Liu et al., 2019 [53]	CRISPR/Cas9
*ZmPLA1*/*NLD*/*MTL*	*Triticum aestivum*	10.0–31.6%	Maternal	Liu et al., 2020 [54]	CRISPR/Cas9
*ZmPLA1*/*NLD*/*MTL*	*Setaria italica*	2.0–3.0%	Maternal	Cheng et al., 2021 [55]	CRISPR/Cas9
*ZmPLA1*/*NLD*/*MTL*	*Triticum aestivum*	10.2–15.2%	Maternal	Tang et al., 2023 [56]	CRISPR/Cas9
*ZmPLA1*/*NLD*/*MTL*	*Saccharum officinarum*	0.6–1.0%	Maternal	Guo et al., 2024 [57]	CRISPR/Cas9
*DMP*	*Zea mays*	0.1–0.3%	Maternal	Zhong et al., 2019 [58]	CRISPR/Cas9
*DMP*	*Arabidopsis thaliana*	1.0–4.0%	Maternal	Zhong et al., 2020 [59]	CRISPR/Cas9
*DMP*	*Medicago*	0.3–0.8%	Maternal	Wang et al., 2021 [60]	CRISPR/Cas9
*DMP*	*Solanum lycopersicum*	0.5–3.7%	Maternal	Zhong et al., 2021 [61]	CRISPR/Cas9
*DMP*	*Nicotiana tabacum*	1.5–1.8%	Maternal	Zhang et al., 2022 [62]	CRISPR/Cas9
*DMP*	*Brassica napus*	1.0–4.4%	Maternal	Li et al., 2022 [63]	CRISPR/Cas9
*DMP*	*Brassica napus*	1.5–2.4%	Maternal	Zhong et al., 2022 [64]	CRISPR/Cas9
*DMP*	*Brassica oleracea*	0.4–2.4%	Maternal	Zhao et al., 2022 [65]	CRISPR/Cas9
*DMP*	*Citrullus lanatus*	0.5–1.1%	Maternal	Chen et al., 2023 [66]	CRISPR/Cas9
*DMP*	*Cucumis sativus*	0.1–0.4%	Maternal	Yin et al., 2024 [67]	CRISPR/Cas9
*ZmPOD65*	*Zea mays*	0.9–7.7%	Maternal	Jiang et al., 2022 [51]	CRISPR/Cas9
*ZmPLD3*	*Zea mays*	0.9–1.0%	Maternal	Li et al., 2021 [68]	CRISPR/Cas9
*ZmBBM2*	*Zea mays*	0.4–3.6%	Maternal	Qi et al., 2023 [71]	CRISPRa
*pPLAIIγ*	*Arabidopsis thaliana*	1.0–1.1%	Maternal	Jang et al., 2023 [73]	CRISPR/Cas9
*ECS*	*Oryza sativa*	3.1–3.6%	Maternal	Zhang et al., 2023 [74]	CRISPR/Cas9
*CENH3*	*Arabidopsis thaliana*	8.0–25.7%	Paternal	Kuppu et al., 2020 [81]	CRISPR/Cas9
*CENH3*	*Daucus carota*	0.2–1.1%	Paternal	Dunemann et al., 2019 [82]	CRISPR/Cas9
*CENH3*	*Triticum aestivum*	7.0–8.0%	Paternal	Lv et al., 2020 [83]	CRISPR/Cas9
*CENH3*	*Arabidopsis thaliana*	13.6–28.6%	Paternal	Han et al., 2024 [84]	CRISPR/Cas9
*CENH3*	*Arabidopsis thaliana*	1.6–24.8%	Maternal	Han et al., 2024 [84]	CRISPR/Cas9
*CENH3*	*Brassica oleracea*	0.5–1.1%	Paternal	Han et al., 2024 [84]	CRISPR/Cas9

## 4. Integration of Haploid Induction Systems with Gene Editing Tools

The CRISPR/Cas9 system and its variants, including Cas9 nickase, dCas9, Cas12a (Cpf1), Cas13 (RNA-targeting), have revolutionized crop breeding by enabling direct modification of trait-associated genes with unprecedented speed and precision. These genome editing tools can introduce targeted mutations, deletions, insertions, and replacements in plant genomes, offering powerful capabilities for crop improvement. However, several significant challenges limit their widespread application in commercial breeding programs. First, many elite crop varieties and inbred lines that form the backbone of commercial breeding remain recalcitrant to genetic transformation, creating a substantial barrier to implementing genome editing tools directly. This transformation recalcitrance often stems from genotype-specific factors, tissue culture response limitations, and complex genetic backgrounds that resist conventional transformation methods. Furthermore, the conventional plant transformation-based gene editing approach typically produces primary transgenic (T_0_) plants with heterozygous edits at both the transgene insertion sites and the targeted genomic loci. Achieving homozygosity through traditional genetic segregation requires multiple generations of self-pollination and careful selection processes. This process becomes particularly challenging and resource-intensive when working with polyploid crops or when targeting multiple unlinked genes simultaneously. The time and resources required for these multiple rounds of selection and breeding can significantly delay the development of improved varieties [86]. To address these limitations, gene editing through haploid induction systems has emerged as a promising alternative strategy. This approach offers several distinct advantages: it enables the production of homozygous genotypes in a single generation, simplifies genome modification in polyploid species by reducing the number of alleles requiring modification by half, and ensures early fixation of desired traits for improved genetic stability. Additionally, transformation of haploid inducer lines that then deliver editing reagents to elite lines without requiring their transformation (HI-Edit/IMGE systems) overcomes the recalcitrance of many elite cultivars to conventional transformation methods. This is particularly valuable for polyploid crops like wheat, where conventional breeding approaches require managing multiple alleles across different genomes. Recent advances in this field include the development of a functional microspore-based gene editing system for wheat [87]. This system employs the Neon transfection system with optimized parameters to directly deliver CRISPR/Cas9 components into wheat microspores. The effectiveness of this approach was demonstrated through successful targeted modifications of both an exogenous DsRed reporter gene and two endogenous wheat genes, *TaLox2* and *TaUbiL1* [87]. In another independent study, introducing target mutations at the wheat *IPK1* locus into isolated wheat microspore cells was achieved through transfection of microspores with ZFN proteins and cell-penetrating peptide complexes [88]. While these pioneering studies have not yet resulted in the successful regeneration of edited haploid plants or DH lines, they have demonstrated the technical feasibility of introducing targeted mutations into isolated wheat microspore cells using both ZFN and CRISPR/Cas9 systems. These advances lay the groundwork for developing more efficient protocols for producing edited haploid plants, which could significantly accelerate crop breeding programs by combining the advantages of DH technology with precise genome editing capabilities. Future developments in this field may focus on optimizing delivery methods, improving tissue culture protocols for regeneration, and developing more efficient editing systems specifically designed for haploid cells.

The unique molecular mechanisms of chromosome elimination during haploid induction have inspired an innovative approach to genome editing that bypasses traditional tissue culture methods. This breakthrough came from understanding how haploid inducer lines selectively eliminate chromosomes in the embryo following double fertilization. This knowledge led to the development of a streamlined platform for delivering genome editing tools directly to embryos through haploid inducer lines. In a remarkable coincidence of scientific advancement, two independent research groups simultaneously developed similar approaches combining haploid induction with CRISPR/Cas9-mediated genome editing. These methods enable genome editing to occur during haploid formation through a single crossing event. The approach developed by the Syngenta research group was termed HI-Edit [89], while the parallel method was named Haploid-Inducer Mediated Genome Editing (IMGE) [90]. Both systems share a common fundamental strategy: they integrate a transgenic CRISPR-Cas cassette into a transformable haploid inducer line, which is then crossed with an elite cultivar.

The process works through several key steps (Figure 1B): (1) The haploid inducer line carrying the CRISPR-Cas system is crossed with the target elite cultivar; (2) during early embryo development, the haploid inducer’s genome (including the CRISPR-Cas transgene) is eliminated; (3) the resulting haploid progeny inherit only the edited genome of the elite cultivar; (4) these haploids are genotyped to identify successful CRISPR-Cas-induced mutations; (5) chromosome doubling is performed to produce homozygous, diploid edited plants.

The integration of haploid inducer-mediated editing systems represents a transformative breakthrough in crop improvement. This innovative approach enables researchers to directly modify genomic targets in elite crop varieties without the traditional constraints of tissue culture or transformation procedures. By eliminating these technical barriers, the technology has made genome editing more accessible and efficient, particularly benefiting work with transformation-recalcitrant elite lines. The system’s ability to produce homozygous edited plants in a single generation significantly accelerates the breeding timeline, while its transgene-free nature simplifies regulatory compliance. This technological advancement has particularly profound implications for breeding programs with limited access to tissue culture facilities and for crops that have historically been difficult to transform. The methodology’s efficiency and accessibility have democratized genome editing capabilities across different research environments and crop species. Looking ahead, ongoing research aims to enhance editing efficiency, broaden the technology’s applicability across diverse crop species, and develop more sophisticated editing capabilities. These future developments may include simultaneous modification of multiple genetic targets and implementation of precise sequence replacement strategies, further expanding the toolkit available to plant breeders and researchers [89,90].

In the study of HI-Edit technology, researchers first explored the capability of *matl* pollen carrying genome editing tools to modify commercial inbred germplasm. They introduced a CRISPR/Cas9 system targeting *MATL* into NP2222, a non-haploid inducer inbred maize line. When pollen from the resulting *matl* mutants (actively expressing Cas9 and gRNA) was used to pollinate commercial inbred line ears, it successfully edited the maternal *MATL* gene post-fertilization. This process generated haploid progeny containing new matl mutations while excluding the male parent genome, demonstrating that *matl*-mediated HI, like CENH3-induced HI, results in post-fertilization elimination of the haploid inducer genome [91]. To assess HI-Edit’s practical breeding applications, researchers transformed NP2222 with constructs expressing Cas9 and gRNAs targeting yield-related genes (*VLHP1*, *VLHP2*, *ZmGW2-1*, and *ZmGW2-2*). These transformed plants were crossed with the RWKS haploid-inducer line, and F2 plants carrying both homozygous matl and the CRISPR-Cas cassette were selected. These plants were then outcrossed to various field-corn inbred lines. Through genetic assays and visual screening, researchers found that HI-Edit produced over 3% edited haploids in five out of six inbred lines tested [89]. Given that the *matl* HI system is limited to monocots due to the absence of clear dicot orthologs, the researchers evaluated HI-Edit in Arabidopsis using the CENH3 HI system. The authors further evaluated HI-Edit within the CENH3 HI system in *Arabidopsis*. They generated an *Arabidopsis* paternal haploid inducer line with the native AtCENH3 replaced with a *ZmCENH3* transgene. The HI line was then transformed with a construct expressing Cas9 and two gRNAs targeting *GLABROUS1* (*GL1*), controlling trichome development. Flowers from confirmed positive transformants that had mutations in *GL1* were crossed by pollen from wild-type *Landsberg erecta* (Ler) for HI-Editing. Analyses of haploid progenies showed high maternal HI-Edit efficiency (16.9%) and faithful inheritance of edited loci in the next generation. Additionally, the intergeneric wide-cross HI system, involving pollination of wheat with maize pollen, was employed to implement HI-Edit in wheat. In this process, NP2222 maize was transformed with a vector expressing *Cas9* and a gRNA targeting putative wheat *GRASSY TILLER1* orthologs *TaGT1-4A*, *TaGT1-4B*, and *TaGT1-4D*. Pollen from the transformed maize plants was then applied to emasculated AC Nanda or cytoplasmic-male-sterile (CMS) wheat spikes. Although the initial editing rate was low, the efficiency of HI-Edit in wheat improved significantly when Cas9 expression in maize was driven by pollen-specific promoters.

For the IMGE approach, a CRISPR/Cas9 construct targeting the *ZmLG1* gene, a key regulator of leaf angle in maize, was generated and used to transform a maize inbred line. The resulting Cas9-positive transgenic plants were crossed with the inducer line CAU5, creating a modified inducer line carrying the CRISPR/Cas9 vector targeting the *ZmLG1* gene. This modified inducer line was subsequently used as the pollen donor in crosses with the inbred line B73 to produce haploid progeny. Among the 245 haploid plants, 10 plants exhibited the *lg1* mutant phenotype and were confirmed to contain an ∼8.8 kb deletion within the *ZmLG1* region, making the editing efficiency approximately 4.1%. Notably, natural chromosome doubling occurred in three of these *lg1*-haploid plants, successfully yielding DH seeds [90].

In light of the successful use of haploid induction to deliver CRISPR/Cas genome-editing reagents for direct genome editing in elite breeding materials of maize, wheat, and Arabidopsis, research efforts have been continually underway to explore the application of haploid induction-mediated genome editing in other crops and to further optimize these systems. Major advances have been achieved in a work that involves intergeneric pollination of wheat with cas9/guide RNA (gRNA)-transgenic maize to facilitate site-directed mutagenesis in wheat plants. Transgene-free and predominantly homozygous M1 plants were efficiently generated for the wheat genes *BRASSINOSTEROID-INSENSITIVE 1* (*BRI1*) and *SEMI-DWARF 1* (*SD1*), which regulate plant height, regardless of the wheat germplasm background [92]. To determine the potential value of DH inducer-mediated genome editing in *Brassica* crops, an artificial hybrid synthesized *Brassica* allooctaploid haploid inducer line Y3380 was successfully used as a CRISPR/Cas9 carrier to simultaneously induce site-specific mutagenesis in multiple homoeologous gene copies of *FAD2* in *B. oleracea* and *B. napus* without the integration of the Cas9 transgene. The mutated progeny induced by Y3380-CRISPR/Cas9 pollen are already DH plants, eliminating the need for additional artificial chromosome doubling processes [93]. Most recently, a novel in vivo genome editing DH system for *Zea mays* L. has been developed. In this study, the ectopic co-expression of *Zea mays* BABY BOOM and the cyclin D-like gene (*Zm-CYCD2*) within unfertilized egg cells enabled the production of maternally derived diploid embryos in vivo. When combined with gene editing, this in vivo approach allows for the production of mature seeds with maternally derived, gene-edited diploid embryos without requiring in vitro tissue culture methods or the use of chemical chromosome-doubling agents [94]. The widespread adoption of HI-Edit/IMGE systems is anticipated to revolutionize crop breeding, particularly given that most crops are challenging to transform, and transformation efficiency often varies significantly among genotypes. The system’s ability to stack multiple gRNAs within a single CRISPR/Cas cassette enables simultaneous modification of multiple genomic targets, facilitating the efficient accumulation of favorable alleles within elite germplasm. While current editing efficiencies may be suboptimal in certain contexts, ongoing advances in genome editing technologies and continued refinement of haploid induction systems hold promise for enhanced performance. These improvements, coupled with the system’s inherent advantages in bypassing traditional transformation barriers, position HI-Edit/IMGE as a powerful tool for accelerating crop improvement programs across diverse species.

## 5. Future Perspectives

The growing demand for global food security necessitates the acceleration of crop breeding, particularly in major staple crops. DH technology offers a powerful solution to this challenge by enabling the production of homozygous plants with stable traits in a single generation. Recent advances in the development of in vivo haploid induction systems and haploid inducer lines mediated genome editing have unlocked new opportunities for integrating genome editing and DH techniques into modern crop breeding programs. Nevertheless, challenges and limitations remain in translating these advancements to practical crop breeding applications.

Low HIR and limited applicability across species remain critical bottlenecks in implementing DH and haploid inducer line-mediated genome editing techniques in crop breeding programs. While crops like maize, wheat, and rice have achieved HIR levels approaching 10% through genome editing of induction-controlling factors, most other crops maintain HIR levels too low for practical breeding applications despite successful haploid inducer system establishment. Recent advances have shown promising strategies to enhance HIR, such as combining different induction mechanisms within a single organism. For instance, integrating tail-altered CENH3 with *ZmMATL*/*PLA1*/*NLD* in *Stock6*-derived lines achieved a maternal HIR of 16.3%, marking a 6.1% improvement over original Stock6 lines [85]. Key genes identified in haploid induction, including *MTL*/*NLD*/*ZmPLA1*, *DMP*, *ZmPOD65*, *CENH3*, and *ECS1*/*2*, present opportunities for multiplex genome editing to synergistically enhance induction efficiency.

A significant challenge in DH technology lies in converting haploids into DHs. The conventional approach requires artificial chromosome doubling in haploid plantlets using mitotic inhibitors, particularly colchicine. While effective, colchicine treatment is labor-intensive, costly, and poses risks to both human health and the environment. The compound’s toxicity often leads to plant losses during application [95]. Although spontaneous chromosome doubling occurs naturally, its frequency varies among plant species and genotypes. A promising alternative combines in planta haploid induction with spontaneous chromosome doubling, potentially eliminating the need for chemical doubling agents. Recent advances include a novel in vivo genome-editing DH system in maize, which successfully integrated haploid induction, spontaneous chromosome doubling, and gene editing to efficiently generate gene-edited maize DH populations [94]. When utilizing CRISPR/Cas-based genome editing in haploid systems, the potential impact on off-target effects presents unique considerations. In haploid cells, off-target mutations have distinct detection and inheritance characteristics compared to diploid systems. With only a single genome copy present, off-target modifications in haploid cells are immediately exposed without being masked by wild-type alleles. This characteristic offers advantages for detection, as sequencing approaches can more easily identify mutations without the complication of heterozygosity. However, it also means that deleterious off-target effects cannot be compensated by a functional allele, potentially increasing their phenotypic impact. Moving forward, research should prioritize identifying the key genes and molecular pathways governing spontaneous chromosome doubling. Enhancing this trait through genome-editing tools like CRISPR/Cas9, while integrating it with DH technology and haploid inducer line-mediated genome editing, will be crucial for improving DH technology efficiency across diverse crops and eliminating dependence on artificial chromosome doubling.

The continued development of genome editing through haploid induction systems offers significant promise for advancing plant breeding. However, the use of haploid inducer lines for the transient expression of genome editing components enables targeted modifications without stable transgene integration in the edited progeny. A major limitation of this approach is that the haploid inducer lines themselves must first be genetically transformed to carry the genome editing machinery. This requirement poses a considerable challenge for crop species that are recalcitrant to genetic transformation, limiting the broader applicability of this method. To overcome this bottleneck, future research should focus on transgene-free genome editing approaches that bypass the need for transformation. Notable strategies include the direct delivery of genome editing reagents—such as CRISPR/Cas9 ribonucleoproteins (RNPs), messenger RNAs, or viral vectors—into reproductive cells or embryos using methods like biolistics, electroporation, or nanoparticle-mediated transfer [96,97]. These non-transgenic delivery systems offer the potential to produce precisely edited plants without introducing foreign DNA, streamlining regulatory approval and increasing public acceptance. Advancing such technologies will be key to expanding the utility of haploid induction-mediated genome editing across a wider range of crops.

## 6. Conclusions

The integration of haploid induction with genome editing technologies represents a transformative advancement in plant breeding, offering solutions to long-standing challenges in crop improvement. This combined approach addresses several critical limitations in traditional breeding methods, particularly the difficulty of transforming recalcitrant elite varieties. By enabling direct genome modification in elite germplasm without conventional transformation procedures, these systems significantly streamline the breeding process. Furthermore, the technology’s capacity for multiplexed editing through stacked gRNAs in single CRISPR/Cas cassettes provides an efficient route to pyramid multiple beneficial traits in elite backgrounds. The versatility of these systems across different crop species highlights their broad potential in agriculture. However, several challenges remain to be addressed for their optimal implementation. Current limitations include variable editing efficiencies across different genetic backgrounds and target sites, necessitating continued optimization of both genome editing tools and haploid induction mechanisms. The development of enhanced editing technologies specifically designed for haploid systems could significantly improve efficiency and precision.

Looking ahead, the field holds promising opportunities for advancement. Research priorities should focus on refining haploid induction mechanisms across diverse crop species and developing more sophisticated editing capabilities. The integration of these systems with other breeding technologies could maximize genetic gain and accelerate crop improvement. As these technologies continue to evolve, their ability to bypass traditional transformation barriers while enabling precise genetic modifications positions them as essential tools in modern plant breeding, potentially revolutionizing how we develop improved crop varieties for future agricultural challenges.

## Figures and Tables

**Figure 1 ijms-26-04779-f001:**
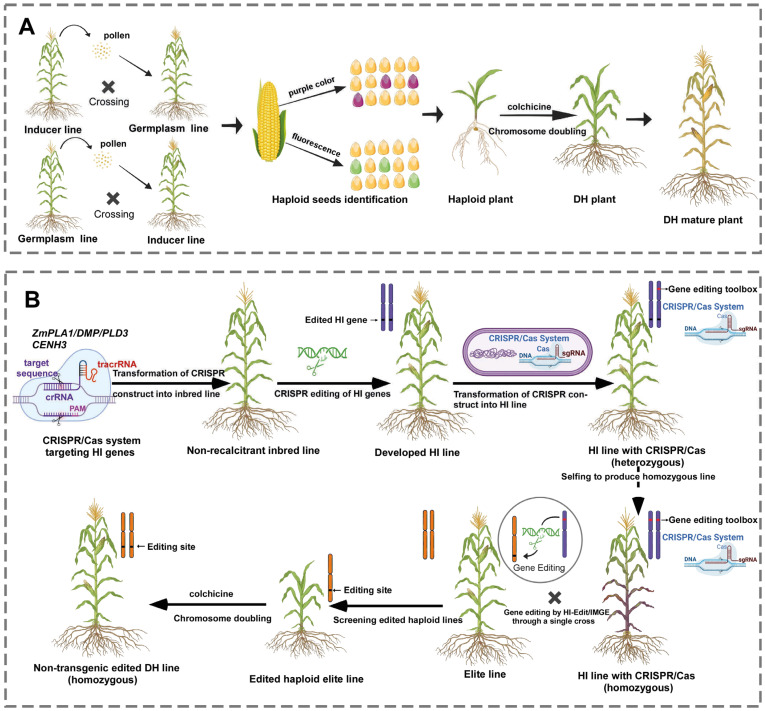
Overview of in vivo haploid induction in DH technology and haploid-inducer-mediated genome editing. (**A**) Technical steps of producing DH lines through HI line-mediated in vivo haploid induction. Haploid inducer lines crossed with donor germplasm trigger the development of haploids that inherit only the maternal/ paternal genome. The resulting haploid seeds are then identified—often through anthocyanin markers or molecular tools—and grow as haploid seedlings. To produce fertile, homozygous DH lines, the haploid plants undergo chromosome doubling, traditionally using mitotic inhibitors such as colchicine. (**B**) The procedures of DH line development via genome editing of HI genes and followed by haploid-inducer-mediated genome editing in maize. A CRISPR/Cas9 cassette targeting HI genes is first transformed into a non-recalcitrant inbred line to generate HI line. Homozygous HI line is then transformed with another genome editing toolbox/cassette to generate HI line carrying the CRISPR/Cas construct. After selfing, a homozygous CRISPR/Cas HI line is obtained. Pollen from this line is then used to pollinate an elite line, resulting in edited haploid plants. These edited haploid plants undergo chromosome doubling to generate non-transgenic, edited DH lines. Figures are created with BioRender.com (https://biorender.com/).

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
