# Peer review of "Advances in Genome Editing Through Haploid Induction Systems"

_ijms, 2025, doi:10.3390/ijms26104779_

Round 1

Reviewer 1 Report

Comments and Suggestions for Authors

This review summarizes the research progress of haploid induction systems in the field of breeding. It discusses the identification and utilization of haploid genes, as well as their integration with gene editing technologies. However, there are still some issues with the structure and presentation of the content. The following issues need to be addressed.

Major comments:

  1. The section on paternal haploid induction genes and related content is too limited to warrant a separate chapter. Instead, the structure of the main text should be reorganized to focus on the mechanisms of action of various haploid induction genes. These mechanisms should be illustrated with diagrams, and the corresponding processes should be explained in detail. This will help readers better understand how haploids are generated and how editing occurs in haploid offspring. Additionally, the haploid induction gene KPL in Arabidopsis should not be overlooked.

Minor comments:

  1. The current title is ambiguous and could lead to misinterpretation. It is recommended to adjust the wording and structure for clarity. Additionally, since the methods and processes related to haploid doubling are only briefly discussed in the manuscript, it is not advisable to include the term "double" in the title. A more focused and precise title would better reflect the main content of the study.

  1. The emphasis on interspecies induction in Figures 1 and 2 should be reduced, as it is less relevant in practical applications. Instead, the figures should highlight the relationship between gene editing and haploid induction. In Figure 2, it should be supplemented with information on gene editing in the induction lines, followed by their use as paternal parents for hybridization with other germplasms. Additionally, the figure captions are too general and do not provide sufficient meaningful information.

  1. In Table 1, The haploid induction rates of CENH3 in Arabidopsis should be distinguished between paternal and maternal origins, and the relevant original references should be cited. Plant names should not be presented using both common names and Latin names simultaneously, they need to be standardized. Additionally, gene names and Latin names should be italicized throughout the manuscript. Please carefully review the entire text to ensure these formatting requirements are met.

  1. While traditional breeding methods require up to eight generations of crossing and selection to stabilize genetic backgrounds and achieve near-complete ho-mozygosity, DH technology offers the most efficient approach by producing completely homozygous lines in just one generation.

Traditional breeding is not all eight generations. To prevent any potential misunderstandings, it is advisable to re-examine the pertinent data.

  1. Given these advantages, DH technology has become a standard tool in major crop improvement programs, particularly for hybrid seed development.

The meaning expressed in this sentence lacks sufficient precision, it is advisable to reorganize the content.

  1. Reference 8 has been available for 20 years, while Reference 9 has been published for 15 years. Therefore, it is not suitable to rely solely on these sources to depict the current situation.

  1. The 0 of T0 needs to be set as a subscript.

Author Response

Response to Reviewer 1

This review summarizes the research progress of haploid induction systems in the field of breeding. It discusses the identification and utilization of haploid genes, as well as their integration with gene editing technologies. However, there are still some issues with the structure and presentation of the content. The following issues need to be addressed.

 Major comments:

  1. The section on paternal haploid induction genes and related content is too limited to warrant a separate chapter. Instead, the structure of the main text should be reorganized to focus on the mechanisms of action of various haploid induction genes. These mechanisms should be illustrated with diagrams, and the corresponding processes should be explained in detail. This will help readers better understand how haploids are generated and how editing occurs in haploid offspring. Additionally, the haploid induction gene KPL in Arabidopsis should not be overlooked.

Response: We appreciate the reviewer’s suggestion. We have restructured the manuscript to focus on the mechanisms of action of various haploid induction genes rather than separating them into maternal and paternal categories. We have combined the content on paternal haploid induction genes with the maternal section to create a more comprehensive discussion of haploid induction mechanisms. We have also added information about the KPL gene in Arabidopsis as suggested (line 288 to line 298).

 Minor comments:

  1. The current title is ambiguous and could lead to misinterpretation. It is recommended to adjust the wording and structure for clarity. Additionally, since the methods and processes related to haploid doubling are only briefly discussed in the manuscript, it is not advisable to include the term "double" in the title. A more focused and precise title would better reflect the main content of the study.

Response: We thank the reviewer for this observation. We have revised the title to be more precise: "Advances in Genome Editing Through Haploid Induction Systems." This title better reflects the main content of the study and removes the term "double" as suggested.

       2. The emphasis on interspecies induction in Figures 1 and 2 should be reduced, as it is less relevant in practical applications. Instead, the figures should highlight the relationship between gene editing and haploid induction. In Figure 2, it should be supplemented with information on gene editing in the induction lines, followed by their use as paternal parents for hybridization with other germplasms. Additionally, the figure captions are too general and do not provide sufficient meaningful information.

Response: Thank you for the suggestion. We have removed the content related to interspecies induction from Figures 1 and 2, and combined them into a single figure to better highlight the relationship between gene editing and haploid induction as recommended. The new Figure 1B (previously Figure 2) now includes information on gene editing in induction lines and their subsequent use as paternal parents for hybridization with other germplasms. We have also revised the figure captions to provide more detailed and meaningful descriptions of the processes illustrated

      3. In Table 1, The haploid induction rates of CENH3 in Arabidopsis should be distinguished between paternal and maternal origins, and the relevant original references should be cited. Plant names should not be presented using both common names and Latin names simultaneously, they need to be standardized. Additionally, gene names and Latin names should be italicized throughout the manuscript. Please carefully review the entire text to ensure these formatting requirements are met.

Response: Thank you for your detailed and constructive feedback. We have revised Table 1 to clearly distinguish between paternal and maternal origins for CENH3-based haploid induction rates in Arabidopsis. To ensure consistency throughout the manuscript, all plant names in Table 1 have been standardized using Latin nomenclature. Additionally, all gene names and Latin names have been italicized throughout the text as suggested. 

      4. While traditional breeding methods require up to eight generations of crossing and selection to stabilize genetic backgrounds and achieve near-complete ho-mozygosity, DH technology offers the most efficient approach by producing completely homozygous lines in just one generation. Traditional breeding is not all eight generations. To prevent any potential misunderstandings, it is advisable to re-examine the pertinent data.

Response: We have re-examined and corrected this statement as: "While traditional plant breeding methods typically require 4-6 successive generations of crossing and selection to stabilize genetic backgrounds and achieve near-complete homozygosity, a process that can span more than a decade depending on the crop and breeding strategy, DH technology offers a dramatically more efficient approach by producing completely homozygous lines in just one generation." (Line 51 to line 55)

     5. Given these advantages, DH technology has become a standard tool in major crop improvement programs, particularly for hybrid seed development. The meaning expressed in this sentence lacks sufficient precision, it is advisable to reorganize the content.

Response: Thank you for the suggestion. We have rephrased this sentence as following: “Given its ability to rapidly produce completely homozygous lines, DH technology has become a widely adopted tool in crop improvement programs, particularly in the development of parental lines for hybrid seed production.” (Line 55 to line 58).

        6. Reference 8 has been available for 20 years, while Reference 9 has been published for 15 years. Therefore, it is not suitable to rely solely on these sources to depict the current situation.

Response: We have updated our references to include more recent sources that better reflect the current state of the field.

       7. The 0 of T0 needs to be set as a subscript.

Response: We have corrected the formatting so that Tâ‚€ appears with the "0" as a subscript throughout the manuscript.

Reviewer 2 Report

Comments and Suggestions for Authors

General comments

The article lacks line numbers so it is difficult to pinpoint the location of errors. The article also needs major revisions for many points mentioned below. The manuscript requires better clarity and accuracy in discussing emerging tools, as some mentioned technologies have been in use for decades. The authors mostly focus on Cas9 of CRISPR; however, it was introduced as a first tool for genome editing. Additionally, newer tools like nickase for SSB, variants of Cas enzymes and RNA editing should also be considered. Construct or reagent delivery challenges should be emphasized, as they remain a major limitation for many crops. The discussion on haploid induction systems should focus more on specific genes and CRISPR tools while reducing excessive details on DH. A table listing target genes, mutation types, and editing tools is necessary, alongside deeper insights instead of just compiling past studies. Finally, the conclusion and future perspectives should be presented separately, with future directions discussing transgene-free genome editing strategies.

Comments for the third paragraph in the introduction section:

  1. But some of these are not emerging and novel since they were introduced decades ago! Also, MAS is not a BT tool; it may be a Genomics one
  2. “traits while significantly reducing costs.” May not be true for all types of editing and for all crops. Many crops, like millets, are still lagging far behind for this tool.
  3. “followed by the introduction of double-strand breaks (DSBs)” But SSB and RNA editing are also common these days.
  4. “Several major genome-editing tools have been developed, including…” Are these all major tools? are you sure?
  5. “…has revolutionized genetic engineering by simplifying…” gene editing!

Comments for the fourth paragraph in the introduction section:

  1. “Continuous innovations are optimizing CRISPR/Cas systems in plants to address…” But as mentioned above, construction delivery is still a challenge for many crops.
  2. Comments for “Haploid Induction Systems in Crop Breeding”: Too much information on DH should be reduced with more focus on specific genes and CRISPR tools
  3. Comments for Maternal Haploid Genes discovery and Genome editing applications for maternal haploid inducer line development. Authors should present a table with names of target genes.
  4. “Gene function validation using CRISPR/Cas9-mediated genome editing and TALEN technology knocked out or mutated the phospholipase gene in a non-inducer background, leading to an average haploid induction rate (HIR) of 2%–6.65% (Table 1).” What type of mutation? what tool was used?
  5. Maternal Haploid Genes discovery and Genome editing applications for maternal haploid inducer line development.” The contents of this section are merely a compilation of past works without any insights. The authors need to write critical insights about each gene.

Comments for the Paternal Haploid Inducer Development via Genome Editing section:

  1. Kindly use only one format for CRISPR/Cas9 (or CRISPR-Cas9)
  2. “Restored frameshift (RFS) mutation in the N-terminal region of Ta-CENH3α-A was introduced via CRISPR-Cas-mediated genome editing.” Is it Cas9 or other tool? Since CRISPR has too many variants of tools

Comments for table 1: Also include one column on type of gene editing tool used

Comments for the integration of Haploid Induction Systems with Gene Editing Tools section:

  1. “The CRISPR/Cas9 system and its variants have revolutionized…” Explain those variants of CRISPR/Cas9.
  2. “Furthermore, the conventional plant transformation-based gene editing approach……promising alternative strategy.” This part is not clear! Authors propose that "the conventional plant transformation-based gene editing approach typically produces primary transgenic (T0) plants with heterozygous edits at both the transgene insertion sites and the targeted genomic loci." Then they state that "to address these limitations, direct editing of gametes or haploid embryos using CRISPR-Cas9 and related genome editing tools has emerged as a promising alternative strategy." But again, is the same transformation required? What is conventional transformation with genome editing?
  3. “Recent advances in this field include the development of a functional microspore-based gene editing system for wheat.” Here and other places, kindly add appropriate references for many sentences
  4. Here and elsewhere, authors stress on single tool, Cas9, which may not be true for several studies.
  5. “…using both ZFN and CRISPR/Cas9 systems.” Again, in the same para, Cas9 is not the only system for CRISPR!
  6. “To determine the potential value of doubled haploid….” Add DH instead of double haploid

Comments for the conclusion and future perspective section:

  1. Two sections should be written separately. First, Conclusion, then Future Perspectives
  2. In the future direction, authors should also discuss on ways of creating transgene-free plants by reagent delivery as discussed in https://doi.org/10.3389/fgeed.2021.805317 and https://doi.org/10.1016/j.plaphy.2023.02.030 (cite and discuss such articles)

Author Response

General comments

The article lacks line numbers so it is difficult to pinpoint the location of errors. The article also needs major revisions for many points mentioned below. The manuscript requires better clarity and accuracy in discussing emerging tools, as some mentioned technologies have been in use for decades. The authors mostly focus on Cas9 of CRISPR; however, it was introduced as a first tool for genome editing. Additionally, newer tools like nickase for SSB, variants of Cas enzymes and RNA editing should also be considered. Construct or reagent delivery challenges should be emphasized, as they remain a major limitation for many crops. The discussion on haploid induction systems should focus more on specific genes and CRISPR tools while reducing excessive details on DH. A table listing target genes, mutation types, and editing tools is necessary, alongside deeper insights instead of just compiling past studies. Finally, the conclusion and future perspectives should be presented separately, with future directions discussing transgene-free genome editing strategies.

Response: We appreciate the reviewer's thorough evaluation. In response to the general comments, we have: (1) Added line numbers throughout the manuscript to facilitate review; (2) Improved clarity in discussing emerging tools, more clearly distinguishing between established and truly novel technologies; (Line 63 to line 65); (3) Expanded our discussion of CRISPR technology beyond Cas9 to include variants of Cas enzymes (Line 83 to line 91);  (4) Added a section discussing construct or reagent delivery challenges as a major limitation; (5) Rebalanced the discussion of haploid induction systems to focus more on specific genes and CRISPR tools; (6) Created a comprehensive table listing target genes and editing tools; (7) Separated the conclusion and future perspectives sections.

Comments for the third paragraph in the introduction section:

  1. But some of these are not emerging and novel since they were introduced decades ago! Also, MAS is not a BT tool; it may be a Genomics one.

Response: We have rephrased this sentence as following: “The rapid advancement and increasing integration of biotechnological and genomic tools—such as high-throughput DNA sequencing, marker-assisted selection, genomic selection, plant transgenic methods, and genome editing—have dramatically enhanced crop breeding efficiency in recent years.”

      2. “traits while significantly reducing costs.” May not be true for all types of editing and for all crops. Many crops, like millets, are still lagging far behind for this tool.

Response: Thank you for your comments. Genome editing, as a tool, holds great potential for targeted modification of specific genes, enabling the rapid improvement of agronomic traits while significantly reducing costs.

      3. “followed by the introduction of double-strand breaks (DSBs)” But SSB and RNA editing are also common these days.

Response: We acknowledge that single-strand breaks (SSBs) and RNA editing have become increasingly common and valuable tools in the broader field of genome editing. However, in the specific context of genome editing applied to haploid induction, there have been no reported studies utilizing SSB or RNA editing approaches to date. Therefore, our focus in this manuscript is primarily on double-strand breaks (DSBs)-mediated genome editing, which currently represents the predominant approach in this specialized research area.

       4. “Several major genome-editing tools have been developed, including…” Are these all major tools? are you sure?

Response:  Thank you for your feedback. We have revised the text to include TALENs and clarified our wording accordingly in the revised version of manuscript. (Line 74 to line 76).

      5. “…has revolutionized genetic engineering by simplifying…” gene editing!

Response: The “genetic engineering" is replaced with "gene editing" throughout the revised manuscript as suggested.

Comments for the fourth paragraph in the introduction section:

  1. “Continuous innovations are optimizing CRISPR/Cas systems in plants to address…” But as mentioned above, construction delivery is still a challenge for many crops.

Response: Thank you for your comments. We acknowledge that despite continuous innovations in optimizing CRISPR/Cas systems in plants, many limitations still exist. Among these, construct delivery remains a major challenge for numerous crops and continues to hinder the broader application of genome editing technologies. We have added discussion on this point in the revised Future Perspectives section.

       2. Comments forHaploid Induction Systems in Crop Breeding”: Too much information on DH should be reduced with more focus on specific genes and CRISPR tools

Response: Thank you for your suggestion. We have reduced the detailed information on different haploid induction systems in the DH production process and only focus on haploid inducer line-mediated haploid production. (Line 124 to line 154).

      3. Comments for Maternal Haploid Genes discovery and Genome editing applications for maternal haploid inducer line development. Authors should present a table with names of target genes.

Response: Thank you for your suggestion. The relevant information, including the names of target genes, has been presented in Table 1 of the manuscript.

       4. “Gene function validation using CRISPR/Cas9-mediated genome editing and TALEN technology knocked out or mutated the phospholipase gene in a non-inducer background, leading to an average haploid induction rate (HIR) of 2%–6.65% (Table 1).” What type of mutation? what tool was used?

Response: Thank you for the comment. In response, we would like to clarify that in the first reports of the two studies that identified and functionally validated MATL/NLD/ZmPLA1, one research group used TALEN technology to introduce small deletions. The other research group used CRISPR/Cas9 to generate different mutations, including a 1-bp insertion, an 11-bp deletion, and a 1-bp deletion in the target region. These mutation types and corresponding tools can be found in references Liu et al., 2017 and Kelliher et al., 2017, which are listed in Table 1.

       5.Maternal Haploid Genes discovery and Genome editing applications for maternal haploid inducer line development.” The contents of this section are merely a compilation of past works without any insights. The authors need to write critical insights about each gene.

Response: We appreciate this valuable feedback and agree that our original text lacked sufficient critical insight. In the revised manuscript, we have substantially restructured this section to go beyond merely compiling past research. We now provide critical analysis of each gene's unique role in the haploid induction pathway, including mechanistic insights into how these genes function, their evolutionary conservation across species, and the specific advantages or limitations they present for practical applications in breeding programs.

For example, we have expanded our discussion of MATL/NLD/ZmPLA1 to include analysis of its proposed biochemical mechanism involving phosphatidylcholine accumulation and subsequent ROS production, highlighting how this understanding provides opportunities for further optimization (Line177 to line 207). For DMP genes, we now critically evaluate the significant differences in induction efficiency across species and discuss the evolutionary implications of their conservation in both monocots and dicots, unlike MTL genes (Line 228 to line 242; Line 262 to line 267). For CENH3-based systems, we analyze why the various required modifications present both challenges and opportunities for emerging gene editing technologies (Line 355 to line 381).

This revised approach provides readers with a deeper understanding of the biological significance and practical utility of each gene, rather than simply cataloging past research. We believe these changes substantially improve the manuscript's contribution to the field.

Comments for the Paternal Haploid Inducer Development via Genome Editing section:

  1. Kindly use only one format for CRISPR/Cas9 (or CRISPR-Cas9)

Response: Thank you for the suggestion. We have changed all instances of CRISPR-Cas9 to CRISPR/Cas9 throughout the text to maintain consistency in formatting.

        2. “Restored frameshift (RFS) mutation in the N-terminal region of Ta-CENH3α-A was introduced via CRISPR-Cas-mediated genome editing.” Is it Cas9 or other tool? Since CRISPR has too many variants of tools

Response: The genome editing was performed using CRISPR/Cas9. We have specified this in the revised manuscript.

Comments for table 1: Also include one column on type of gene editing tool used

Response: One column on the type of gene editing tool has been added to Table 1 in the revised manuscript.

Comments for the integration of Haploid Induction Systems with Gene Editing Tools section:

  1. “The CRISPR/Cas9 system and its variants have revolutionized…” Explain those variants of CRISPR/Cas9.I

Response: Thanks for the suggestion. We have listed several major Cas9 variants in the revised manuscript. (Line 385 to line 386).

   2. “Furthermore, the conventional plant transformation-based gene editing approach……promising alternative strategy.” This part is not clear! Authors propose that "the conventional plant transformation-based gene editing approach typically produces primary transgenic (T0) plants with heterozygous edits at both the transgene insertion sites and the targeted genomic loci." Then they state that "to address these limitations, direct editing of gametes or haploid embryos using CRISPR-Cas9 and related genome editing tools has emerged as a promising alternative strategy." But again, is the same transformation required? What is conventional transformation with genome editing?

Response: Thank you for highlighting this lack of clarity. In the revised manuscript, we have completely rewritten this section to better explain the distinction between conventional and alternative approaches.

To clarify: Conventional plant transformation-based gene editing involves delivering CRISPR/Cas9 components into somatic plant cells (usually via Agrobacterium or biolistics), followed by tissue culture, selection, and regeneration of transgenic plants. This approach typically produces primary transgenic (Tâ‚€) plants with heterozygous edits, requiring multiple generations of selfing to obtain homozygous, transgene-free plants. In contrast, the alternative strategy we discuss involves directly delivering editing reagents into gametes or haploid cells/embryos. This approach offers distinct advantages because that edits in haploid cells automatically become homozygous upon chromosome doubling, eliminating the need for multiple generations of selfing. While transformation is still required in some variations of this approach, it's performed on the haploid inducer line rather than the elite variety being modified.

We have clarified in the revised manuscript that there are two main implementations of this concept: (1) Direct transformation of isolated microspores or haploid cells (still requiring tissue culture but resulting in instant homozygosity); (2) Transformation of haploid inducer lines that then deliver editing reagents to elite lines without requiring their transformation (HI-Edit/IMGE systems). These distinctions are now clearly explained in the revised manuscript, with explicit descriptions of the transformation requirements for each approach (Line 403 to line 411).

        3. “Recent advances in this field include the development of a functional microspore-based gene editing system for wheat.” Here and other places, kindly add appropriate references for many sentences

Response: Thank you for the suggestion. We have added a reference for this sentence in the revised manuscript. Additionally, we thoroughly checked the context and included appropriate references throughout.

       4. Here and elsewhere, authors stress on single tool, Cas9, which may not be true for several studies.

Response: Thank you for this important observation. You are correct that our manuscript placed undue emphasis on Cas9 alone, potentially misrepresenting the diversity of CRISPR tools used in this field. In the revised manuscript, we have broadened our discussion to include other CRISPR nucleases and genome editing tools (Line 83 to line 91, line 385 to line 389).

In addition, we have carefully reviewed the entire manuscript to replace instances where we incorrectly limited discussion to Cas9 alone. Where appropriate, we now use broader terminology such as "CRISPR-Cas systems" or specifically mention other nucleases like Cas12a/Cpf1, Cas13, and base editors when discussing research that utilized these tools. We have also updated our table to properly indicate which specific CRISPR variant was used in each study. This more inclusive approach better reflects the current state of the field and acknowledges the important contributions made using various CRISPR nucleases and other genome editing technologies. We appreciate you bringing this issue to our attention.       

        5. “…using both ZFN and CRISPR/Cas9 systems.” Again, in the same para, Cas9 is not the only system for CRISPR!

Response: Thank you for this important observation. You are correct that our manuscript placed undue emphasis on Cas9 alone, potentially misrepresenting the diversity of CRISPR tools used in this field. In the revised manuscript, we have broadened our discussion to include other CRISPR nucleases and genome editing tools (Line 83 to line 91, line 385 to line 389).

In addition, we have carefully reviewed the entire manuscript to replace instances where we incorrectly limited discussion to Cas9 alone. Where appropriate, we now use broader terminology such as "CRISPR-Cas systems" or specifically mention other nucleases like Cas12a/Cpf1, Cas13, and base editors when discussing research that utilized these tools. We have also updated our table to properly indicate which specific CRISPR variant was used in each study. This more inclusive approach better reflects the current state of the field and acknowledges the important contributions made using various CRISPR nucleases and other genome editing technologies. We appreciate you bringing this issue to our attention.       

      6. “To determine the potential value of doubled haploid….” Add DH instead of double haploid

Response: Thank you for the suggestion. We have replaced "doubled haploid" with "DH" in the revised manuscript.

Comments for the conclusion and future perspective section:

  1. Two sections should be written separately. First, Conclusion, then Future Perspectives

Response: We have revised the manuscript by separating the Conclusion and Future Perspectives into two distinct sections as suggested. Thank you for this recommendation.

    2. In the future direction, authors should also discuss on ways of creating transgene-free plants by reagent delivery as discussed in https://doi.org/10.3389/fgeed.2021.805317 and https://doi.org/10.1016/j.plaphy.2023.02.030 (cite and discuss such articles)

Response: We have added a discussion on creating transgene-free plants through reagent delivery and have cited the suggested references in the revised manuscript (Line 605 to line 620). Thank you for this valuable recommendation.

Reviewer 3 Report

Comments and Suggestions for Authors

The manuscript presents a timely and comprehensive review of the intersection between gene editing technologies and haploid/diploid induction systems in plant breeding. The paper effectively conveys the central theme, emphasizing the significance of integrating these induction systems with genome-editing platforms like CRISPR-Cas to enhance breeding efficiency and accelerate the development of homozygous lines.

The subject matter addresses a critical frontier in modern plant biotechnology, aligning with current trends and research priorities.

The paper outlines key areas covered in the review — mechanisms, applications, challenges, and future prospects — providing a structured approach.

The paper has an interdisciplinary approaches - the integration of genome editing with haploid/diploid induction is of interest across molecular biology, genetics, and agronomy.

In my opinion, the topic is highly relevant, especially in the context of growing demands for sustainable agriculture and global food security. By highlighting both the mechanistic understanding and practical applications of haploid and diploid inducers, the paper appears to offer a valuable synthesis for researchers and breeders alike.

  1. What is the main question addressed by the research?

The manuscript aims to explore how haploid and diploid induction systems can be used to enhance genome editing efficiency in plants. It specifically focuses on methods such as haploid induction through in vivo systems and how these systems could be used to rapidly generate homozygous edited lines, which is valuable for functional genomics and in plant breeding.

  1. Is the topic original or relevant to the field? Does it address a specific gap?

In my opinion the topic is both original and highly relevant to current plant genetics and breeding research. The integration of genome editing with doubled haploid (DH) technology is a novel and promising area that addresses a significant gap in speeding up the generation of stable transgenic or gene-edited lines. It adds value by summarizing current approaches and highlighting opportunities to overcome longstanding bottlenecks in plant breeding timelines.

  1. What does it add to the subject area compared with other published material?

The paper presents the news in the field by offering a comparative overview of haploid and diploid induction systems across major crops, effectively linking these approaches with advanced gene-editing tools such as CRISPR/Cas9 to highlight their synergistic potential. It introduces emerging strategies like haploid induction-mediated genome editing (HI-Edit) and the concept of targeted mutations, positioning them as promising tools for enhancing the efficiency of genetic transformation. Furthermore, the authors propose potential frameworks to improve both the precision and throughput of genome editing applications, writing valuable insights for accelerating plant breeding and functional genomics.

  1. What specific improvements should the authors consider regarding the methodology?

Although this is a review paper and does not present original experimental methodology, several enhancements could improve its clarity and depth. The inclusion of a summarized table comparing the efficiency, advantages, and limitations of various haploid and diploid induction systems across different species would provide valuable comparative context. Additionally, a more thorough discussion on off-target effects—particularly how haploid systems might influence the frequency or detection of such effects—would strengthen the section on genome editing precision. Finally, offering greater insight into the biological triggers of haploid induction, such as centromere dysfunction (Wang, S., Jin, W. & Wang, K. Centromere histone H3- and phospholipase-mediated haploid induction in plants. Plant Methods 15, 42 (2019). https://doi.org/10.1186/s13007-019-0429-5), would benefit readers less familiar with the underlying processes and enhance the overall comprehensiveness of the review.

  1. Are the conclusions consistent with the evidence and arguments presented and do they address the main question posed?

In my opinion the conclusions are well-aligned with the body of the review. The authors effectively argue that integrating haploid/diploid induction systems with CRISPR-based genome editing can significantly accelerate crop improvement efforts. They also appropriately note the challenges and open questions that remain, such as species-specific limitations and the need for further optimization.

  1. Are the references appropriate?

Overall, the references cited are current and relevant, reflecting a solid balance between foundational studies and recent advancements, including key publications from 2021 to 2023. The additional references on recent developments in Gamete Targeted Editing (GTE) beyond 2022 can be used in the article.

  1. Additional comments on tables and figures:

The figures effectively illustrate the workflow of HI-Edit systems.

Author Response

The manuscript presents a timely and comprehensive review of the intersection between gene editing technologies and haploid/diploid induction systems in plant breeding. The paper effectively conveys the central theme, emphasizing the significance of integrating these induction systems with genome-editing platforms like CRISPR-Cas to enhance breeding efficiency and accelerate the development of homozygous lines.

The subject matter addresses a critical frontier in modern plant biotechnology, aligning with current trends and research priorities.

The paper outlines key areas covered in the review — mechanisms, applications, challenges, and future prospects — providing a structured approach.

The paper has an interdisciplinary approach - the integration of genome editing with haploid/diploid induction is of interest across molecular biology, genetics, and agronomy.

In my opinion, the topic is highly relevant, especially in the context of growing demands for sustainable agriculture and global food security. By highlighting both the mechanistic understanding and practical applications of haploid and diploid inducers, the paper appears to offer a valuable synthesis for researchers and breeders alike.

Response: We thank Reviewer 3 for their positive evaluation of our manuscript. We are pleased that the reviewer found our paper to be timely, comprehensive, and addressing a critical frontier in modern plant biotechnology.

  1. What is the main question addressed by the research?

The manuscript aims to explore how haploid and diploid induction systems can be used to enhance genome editing efficiency in plants. It specifically focuses on methods such as haploid induction through in vivo systems and how these systems could be used to rapidly generate homozygous edited lines, which is valuable for functional genomics and in plant breeding.

        2. Is the topic original or relevant to the field? Does it address a specific gap?

In my opinion the topic is both original and highly relevant to current plant genetics and breeding research. The integration of genome editing with doubled haploid (DH) technology is a novel and promising area that addresses a significant gap in speeding up the generation of stable transgenic or gene-edited lines. It adds value by summarizing current approaches and highlighting opportunities to overcome longstanding bottlenecks in plant breeding timelines. 

         3. What does it add to the subject area compared with other published material?

The paper presents the news in the field by offering a comparative overview of haploid and diploid induction systems across major crops, effectively linking these approaches with advanced gene-editing tools such as CRISPR/Cas9 to highlight their synergistic potential. It introduces emerging strategies like haploid induction-mediated genome editing (HI-Edit) and the concept of targeted mutations, positioning them as promising tools for enhancing the efficiency of genetic transformation. Furthermore, the authors propose potential frameworks to improve both the precision and throughput of genome editing applications, writing valuable insights for accelerating plant breeding and functional genomics. 

       4. What specific improvements should the authors consider regarding the methodology?

 Although this is a review paper and does not present original experimental methodology, several enhancements could improve its clarity and depth. The inclusion of a summarized table comparing the efficiency, advantages, and limitations of various haploid and diploid induction systems across different species would provide valuable comparative context. Additionally, a more thorough discussion on off-target effects—particularly how haploid systems might influence the frequency or detection of such effects—would strengthen the section on genome editing precision. Finally, offering greater insight into the biological triggers of haploid induction, such as centromere dysfunction (Wang, S., Jin, W. & Wang, K. Centromere histone H3- and phospholipase-mediated haploid induction in plants. Plant Methods 15, 42 (2019). https://doi.org/10.1186/s13007-019-0429-5), would benefit readers less familiar with the underlying processes and enhance the overall comprehensiveness of the review.

Response: Thank you for these valuable suggestions for enhancing our review paper. We have implemented the following improvements: (1) We have expanded our discussion on off-target effects, particularly addressing how haploid systems might influence both the frequency and detection of such effects. In the revised manuscript, we explain how the reduced ploidy in haploid cells can simplify the detection of off-target mutations while potentially increasing their phenotypic impact, providing important considerations for researchers applying these techniques (Line 590 to line 598); (2) We have added a more detailed section on the biological triggers of haploid induction, with particular emphasis on centromere dysfunction. We have cited and incorporated insights from Wang et al. (2019) regarding centromere histone H3- and phospholipase-mediated haploid induction mechanisms (Line 317 to line 345). This addition provides readers with a deeper understanding of the underlying biological processes that enable haploid induction. We believe that these revisions have substantially improved our manuscript and addressed all the concerns raised by the reviewer. We once again thank the reviewer and editors for their valuable feedback and careful consideration of our work.

    5. Are the conclusions consistent with the evidence and arguments presented and do they address the main question posed?

 In my opinion the conclusions are well-aligned with the body of the review. The authors effectively argue that integrating haploid/diploid induction systems with CRISPR-based genome editing can significantly accelerate crop improvement efforts. They also appropriately note the challenges and open questions that remain, such as species-specific limitations and the need for further optimization.

      6. Are the references appropriate?

Overall, the references cited are current and relevant, reflecting a solid balance between foundational studies and recent advancements, including key publications from 2021 to 2023. The additional references on recent developments in Gamete Targeted Editing (GTE) beyond 2022 can be used in the article. 

      7. Additional comments on tables and figures:

The figures effectively illustrate the workflow of HI-Edit systems.

Round 2

Reviewer 1 Report

Comments and Suggestions for Authors

Thank you for addressing my comments. I have no further requests.

Reviewer 2 Report

Comments and Suggestions for Authors

The article can now be accepted for publication.